# Antenatal screenings and maternal diagnosis among pregnant women in Sao Tome & Principe—Missed opportunities to improve neonatal health: A hospital-based study

Alexandra Vasconcelos[1]*, Swasilanne Sousa[2], Nelson Bandeira[2], Marta Alves[3], Ana Luísa Papoila[3], Filomena Pereira[1], Maria Céu Machado[4]

1 Unidade de Clínica Tropical—Global Health and Tropical Medicine (GHTM), Instituto de Higiene e Medicina Tropical (IHMT), Universidade Nova de Lisboa, Lisboa, Portugal, 2 Hospital Dr. Ayres de Menezes, São Tomé, Sao Tome and Principe, 3 CEAUL, NOVA Medical School/Faculdade de Ciências Médicas, Universidade Nova de Lisboa, Lisboa, Portugal, 4 Faculdade de Medicina de Lisboa, Universidade de Lisboa, Lisboa, Portugal

* alexandravasc@gmail.com

## Abstract

Newborn mortality and adverse birth outcomes (ABOs) in Sao Tome & Príncipe (STP) are overwhelmingly high, and access to quality-antenatal care (ANC) is one of the strategies to tackle it. This study aimed to fill the gaps in ANC screenings with a focus on how to improve neonatal outcomes. We conducted a retrospective hospital-based study in which ANC pregnancy cards were reviewed. Screenings were described and compared according to the total number of ANC contacts: 1–3 (inadequate), 4–7 (adequate), and ≥8 (complete). The collected data were entered into QuickTapSurvey and exported to SPSS version 25 for analysis. Statistical significance was considered at a p-value ≤0.05. A total of 511 ANC pregnancy cards were reviewed. Mothers' mean age was 26.6 (SD = 7.1), 51.7% had a first trimester early booking, 14.9% (76) had 1–3 ANC contacts, 46.4% (237) had 4–7 and 38.7% (198) ≥8. Screening absence was found in 24%-41%, lack of money was registered in 36%. Pregnant women had no screening performed for HIV in 4.5%, syphilis in 8.8%, HBV 39.3%, malaria 25.8%, hemoglobin 24.5%, blood glucose 45.4%, urine 29.7%, stool exams 27.8% and 41.1% had no ultrasound. Screening completion for blood group, HIV, malaria, urine, hemoglobin, and coproparasitological exam were found to have a statistically significant difference (p<0.001) for the complete ANC group when compared to other groups. Antenatal problems identified were: 1) bacteriuria (43.2%); 2) maternal anemia (37%); 3) intestinal parasitic infections (59.2%); 4) sickle cell solubility test positive (13%); and 5) a RhD-negative phenotype (5.8%). Missed-ANC treatments were up to 50%. This study reveals a coverage-quality gap in STP since no pregnant woman is left without ANC contact, although most still miss evidence-based screenings with an impact on neonatal outcomes. Strategies such as implementing a total free ANC screening package in STP would enhance maternal diagnosis and prompt treatments.

**Data Availability Statement:** The datasets used and analyzed during the current study are all available within the manuscript itself.

**Funding:** AV was funded by the Fundação para a Ciência e Tecnologia (FCT) (https://www.fct.pt/index.phtml.pt/), grant number SFRH/BD/117037/2016. The funder had no role in the study design, data collection and analysis, decision to publish or preparation of the manuscript.

**Competing interests:** The authors have declared that no competing interests exist.

**Abbreviations:** STP, Sao Tome and Principe; HAM, Hospital Dr. Ayres de Menezes; SDG, Sustainable Development Goal; ANC, antenatal care; MICS, Multiple Indicator Cluster Survey; WHO, World Health Organization; HIV, human immunodeficiency virus; IPTp, Sulfadoxine-Pyrimethamine for intermittent preventive treatment for malaria during pregnancy; ITN, Insecticide Treated Nets; USD, United States Dollar; GA, gestational age.

# Background

Antenatal care (ANC) was a concept created in the early decades of the past century that is now widely acknowledged for its potential for improving newborn survival and health [1–3]. Thus, efforts to tackle low-quality ANC are a key strategy to reduce the high-burden of neonatal morbidity and mortality in sub-Saharan African (SSA) countries [4, 5].

The World Health Organization (WHO) latest recommendations on "antenatal care for a positive pregnancy experience" state that it should start early for timely detection and treatment of maternal problems with a goal of a minimum of eight contacts per pregnant woman [6]. This guideline was published in 2016, and currently, most SSA countries are still in transition from the previous four ANC visits from the 2001 Focused Antenatal Care (FANC) model [2, 7, 8]. In addition, the WHO also preconizes a package of evidence-based screenings for detecting maternal problems and implementing prompt treatments. For instance, interventions such as timely maternal HIV diagnosis can reduce neonatal transmission of HIV to less than 5% with initiation of antiretroviral treatments [9–11]. Additionally, syphilis, when properly treated, prevents small birth weight deliveries in 20% and stillbirths up to 40% [6, 10, 12]. Other ANC screenings are also designed to prevent adverse birth outcomes (ABOs). Anemia in pregnant women, for instance, is another major setback in newborn health, causing different ABOs, such as intrauterine growth retardation, prematurity, low birth weight (LBW), and fetal death [13]. Asymptomatic urinary tract infection in pregnancy, when not treated, is also associated with a 30% risk of developing maternal pyelonephritis, causing ABOs such as LBW and/or preterm delivery [14, 15].

Another example is the prevention of Rh alloimmunization and detection of sickle cell disease (SCD). Alloimmunization prevention reduces perinatal ABOs during a second RhD-positive pregnancy [16]. In its mildest form, the newborn has sensitized red blood cells; on the other hand, the severe form of hemolytic disease may result in intrauterine death or newborn jaundice, anemia, and infant developmental problems [16, 17]. The setback is that antenatal prophylaxis with anti-D immunoglobulin in nonsensitized Rh-negative women at 28 and 34 weeks of gestation is not available in most SSA countries [16]. Similarly, SCD is often undiagnosed in SSA and is associated with complications such as, fetal loss and intrauterine growth restriction [18]. Additionally, the high prevalence of this undiagnosed noncommunicable disease contributes to excess mortality in children under five years of age in Africa [18].

For all the above reasons, ANC is an excellent opportunity to reach pregnant women to prevent neonatal mortality and ABOs, providing a full package with prophylactic medication, vaccines, diagnosis, and treatment, as well as with health education programs [19–23].

Currently, ANC varies tremendously among SSA countries due to factors related to national health policies, screening costs, facilities, and users [24–26]. Additionally, most studies in this field, conducted in SSA contexts, are mainly focused only on the coverage and total number of ANC contacts, disregarding screenings and interventions accomplished [27–29].

Sao Tome & Principe (STP), a SSA Western country, has two islands, with a total land surface of approximately 1,001 $km^2$ (859 $km^2$ for Sao Tome and 142 $km^2$ for Principe) [30–32]. The country has no land borders, but lies relatively close to the coasts of Gabon, Equatorial Guinea, Cameroon, and Nigeria. The country has already reached some of the WHO indicator goals, such as 98% of pregnant women accessing ANC at least once, 95.4% receiving the service of a skilled birth attendant at birth, and 91% obtaining first postnatal care within the first 24 hours after birth [30]. Additionally, there was considerable progress in the country for the maternal mortality ratio, with a reduction from 158 to 74 maternal deaths per 100,000 live births between 2009 and 2014, although neonatal mortality remains a problem [30, 31]. Moreover, ABOs, such as preterm delivery, LBW, small for gestational age, neonatal sepsis, and

congenital anomalies, all amenable to prevention with high-quality ANC, are also a high-burden in STP [22, 32, 33].

In STP, the transition from the model of four ANC visits to the latest WHO recommendation of eight contacts occurred between 2017 and 2018 [32]. Actual ANC free-package in the country includes blood pressure measurement, iron/folic acid supplementation, malaria prevention, tetanus vaccination, and rapid tests (HIV, syphilis, and hepatitis B surface antigen). Blood tests (hemoglobin, glucose, blood group, Rhesus factor, sickle cell, widal test), urine tests, vaginal swabs, coproparasitological exams, and obstetric ultrasounds are requested but have costs for pregnant women [32]. The ANC interventions are documented in an ANC pregnancy card that is carried by all pregnant women to ANC services and to the maternity unit.

Therefore, this study aimed to fill the gaps in ANC—beyond the number of contacts—by identifying screenings performed, maternal problems detected, and treatments provided. To the authors' knowledge, no study on ANC has ever been done in STP. This current study is included in a broader project on neonatal mortality and adverse birth outcomes in Sao Tome & Principe [34–36].

With this study, we intended to contribute to the development of a strategy to improve ANC in STP for neonatal mortality and ABOs high-burden reduction as endorsed in the post-2015 Sustainable Development Goals (target 3.1) [22, 37, 38].

## Material and methods

### Ethics statement

The study was approved and consented to by the Ministry of Health of Sao Tome & Principe and by the main board of Hospital Dr. Ayres de Menezes, since at the time the study protocol was submitted, there was no ethics committee in Sao Tome & Principe. Only recently has the country National Ethics Committee been appointed. Previously, study analysis and approval were performed by the Ministry of Health and the institution where the study was to be conducted, and that is what we have done. All methods were performed in accordance with the relevant guidelines and regulations in practice. Written informed consent was obtained from all participants (or their parent or legal guardian in the case of adolescent under 16 or illiterate participants) after the purpose of the research to review the information documented in the ANC pregnancy card was explained orally by the main investigator. Participation in the survey was voluntary, as participants could decline to participate at any time during the study.

### Study design

This was a retrospective, hospital-based analytical study conducted among pregnant women admitted to the maternity unit of the Hospital Dr. Ayres de Menezes (HAM) for delivery. A total of 511 pregnant women's ANC pregnancy cards were reviewed. This study was purely descriptive in nature as part of the study design.

### Setting

The study was conducted at the maternity unit of Hospital Dr. Ayres de Menezes (HAM), the only hospital in the country. This maternity unit is responsible for the delivery of 82.4% of all newborns in STP, representing a 4.500 annual birth cohort [32]. Therefore, this study setting allows to have a global picture of ANC practices in the country.

Antenatal care services in STP are available to women on all working days and contacts are carried out by nurses [32]. They are normally organized along the five thematic components

of service provision stipulated by the WHO guidelines: 1) history taking, 2) physical examination, 3) laboratory examinations, 4) drug administration and immunization and 5) health education [6, 24]. ANC pregnancy cards are an important working tool. In Sao Tome & Principe, it has ten pages, and health workers use them to follow-up the pregnancy by registering personal information, physical examinations, laboratory tests, treatments, and other notes. Pregnant women carry the ANC pregnancy card to each ANC contact and to the hospital at the time of delivery. Screening of human immunodeficiency virus (Determine test), hepatitis B virus (hepatitis B surface antigen—HBsAg) and syphilis (rapid plasma reagin test) are included in the STP´s essential package of interventions for maternal and newborn health, which is done using an opt-out approach and free of charge [23]. Other essential laboratory tests, rapid tests and ultrasounds have costs for pregnant women. Each woman pays approximately 17 United States Dollar (USD) for all mandatory tests (blood, urine, feces, gynecological exams) and obstetric ultrasounds [32].

Midstream urine culture (the gold standard) is the recommended method for diagnosing asymptomatic bacteriuria, but urine culture is not available in STP, so urine dipstick tests in health posts and laboratory urine in other centers are the methods used [6, 32].

Full blood count testing for diagnosing anemia in pregnancy is available only at HAM [8, 13, 32]. Other ANC services perform on-site hemoglobin testing using a hemoglobinometer.

Blood culture is commonly used as the reference standard for typhoid diagnosis but requires sophisticated equipment not readily available in most SSAs, as in STP [39–41]. The Widal test is a quantitative agglutination test that identifies serum antibodies against Salmonella antigens O-somatic and H-flagellar, and in STP, it is performed once during pregnancy [41]. The test is considered positive if a convalescent phase serum sample has a fourfold higher titter than an acute sample [41].

A sickle cell solubility test, which involves treating a thin blood film with sodium dithionate under hypoxic conditions and observing for sickling under a light microscope, is the screening technique available in STP and is performed on pregnant women with anemia or clinical suspicion [42, 43]. A positive result can suggest either sickle cell anemia or the sickle cell trait [42]. The diagnosis of sickle cell disease cannot be established with certainty by means of the sickling test alone but must be substantiated by electrophoretic analysis, unfortunately not available in the country [32].

## Participants

All pregnant women admitted to the HAM maternity unit for delivery constituted the source population, whereas the study populations were selected pregnant women admitted to the HAM maternity unit during the study period. The recruitment occurred from July 2016 to November 2018.

The eligibility criteria for participants were as follows: 1) all women admitted to the hospital for delivery with a gestational age of 28 weeks or more and 2) women who gave birth outside the hospital but were later admitted at HAM on the day of birth.

The exclusion criteria included the following: 1) women with induced termination of pregnancy for medical reasons, 2) adolescent or illiterate mothers who had not obtained permission from their parents or legal guardians to participate in the study, and 3) women without an ANC pregnancy card.

Of the 534 eligible participants, 16 women were excluded, nine due to not having an ANC pregnancy card and seven due to missing information on the ANC regarding the number of ANC contacts. A total of five hundred and eleven pregnant women were included in this study.

## Sampling method

The software used for sample calculation was Raosoft (http://www.raosoft.com/samplesize.html), and the value sample was also supported by PASS software (https://www.ncss.com/software/pass/). A minimum sample size of S = 355 was recommended, which placed the right dimension between 355 (95%) and 579 (99%) confidence. For the original study, participants were enrolled based on the following assumptions: two-sided 95% confidence level, power of 80% to detect an odds ratio of at least 2 for adverse birth outcomes. Since the sample size was not calculated for present outcomes, a power analysis was performed, varying from 77% to 87% for outcomes such as having four ANC contacts for this study.

Pregnant women were invited to participate in the study after admission to the maternity unit. Invitations were made by the investigator (AV) and occurred during daytime hours, from Monday to Friday. Participants were selected randomly until the required sample size was achieved. Each morning, from the pile of pregnant women admission folders, every second interval folder was selected and then carried on asking for consent for enrollment before starting the data collection.

## Data sources

The ANC pregnancy card of each study participant was used to collect personal data, obstetric and pregnancy information as well as physical examinations, laboratory tests, and treatments.

## Study variables

The information was collected from ANC pregnancy cards on mothers' age, place of residence, educational status, occupation, and obstetrics characteristics such as: parity, gravidity, history of abortion and stillbirth. Gravidity was categorized as primigravida (1), multigravida (2–4) and grandmultigravida (above 5). Parity was classified as nulliparous (0), multiparous (1–4) and grandmultigravida (above 5).

Employment was defined for those who engaged in one economic activity and not employed for those who did not engage in economic activities. Residence was grouped into urban and rural areas. The urban area was living in the Água Grande district and rural areas in Mé-Zochi, Cantagalo, Lobata, Lembá, Cáué and Principe Island.

For ANC characterization, the following variables were included: timing at first ANC booking on the first (conception to 13th gestational age week), second (14–26 week) or third trimester (27–40 week), number of ANC contacts, high-risk pregnancy scoring notification, screenings, interventions, diagnosis, and treatments. Reasons registered by the nurses in the ANC pregnancy cards justifying why the screenings were not performed were also collected. The main evidence-based interventions were grouped, described, and compared according to the number of ANC contacts per pregnant woman: 1–3 (inadequate), 4 to 7 visits (adequate) and 8 or more (complete) [22–25].

## Bias

As this was a retrospective study, the reasons for the poorly recorded information were not explored, and only the completeness of the recording could be analyzed.

## Data management and quality

Data were secured in a confidential and private location. Participants were referred to by identification numbers, and the informed consent forms were kept separate. Both could only be linked by a coding sheet available to the investigator. Supervision of data collection and

continuous follow-up were made by the supervisors. The investigator (a pediatrician) executed and was responsible for field study activities as follows: obtaining consent and enrollment of the pregnant women, data collection from ANC pregnancy cards, and entry into the database.

## Statistical methods

ANC characterization (screenings, diagnosis, and treatments) and characteristics of study participants were described with frequencies and percentages or mean (standard deviation) and range (min-max), as appropriate. Comparisons of screenings between the different groups (inadequate ANC, adequate and complete ANC) were assessed using the Chi-squared test or Fisher's exact test. A level of significance $\alpha = 0.05$ was considered. Data collected from the ANC pregnancy card were directly entered into QuickTapSurvey (2010–2021 Formstack) an offline survey app tool, and further analysis was conducted in the Statistical Package for the Social Sciences for Windows, version 25.0 (IBM Corp. Released 2017. IBM SPSS Statistics for Windows, Version 25.0. Armonk, NY: IBM Corp.).

## Results

### Sociodemographic characteristics of the study participants

A total of 511 pregnant women´s ANC pregnancy cards were reviewed. The mother´s mean age was 26.6 (7.1) years old, between 14 and 43 years of age. Twenty-eight (5.5%) were aged between 14 and 16, 14.7% (75) 17–19, 63.2% (323) 20–34 years and 16.6% (85) were older than 35 years of age. Regarding mothers´ education, 57.9% (296) had primary education, and 32.1% (164) had secondary education. Higher education was found in 6.5% (33). Illiteracy was found in 3.5% (18). Concerning employment status, 70.6% (361) were not employed, and 28.8% (147) were employed. Urban area living was confirmed in 44.6% (228), while 53.6% (274) were living in rural areas, such as Mé-Zochi 28.4% (146), Cantagalo 11.4% (58), Lobata 6.3% (32), Lembá 4.3% (22), Caué 1.8% (9) and Principe Island 1.4% (7).

### Antenatal care (ANC) characterization

**ANC: Timing and number of contacts.** Early first ANC contact during the first trimester of gestation was reached by 51.7% (268/446) of pregnant women. Late booking with a first ANC contact during the second and third trimesters occurred in 31.5% (163/446) and 2.9% (15/446) of participants, respectively. Data regarding the timing of the first ANC attendance were missing from 65 (14.6%) ANC pregnancy cards. The detailed distribution of gestational age in weeks and by trimester at the initiation of ANC contact is shown in Fig 1. Regarding the total number of ANC contacts, 14.9% (76) had fewer than 4 ANC contacts, 46.4% (237) had 4 to 7 and 38.7% (198) had complete ANC ($\geq$8 contacts) as described in Table 1.

**ANC: High-risk pregnancy scoring.** Notification of a high-risk pregnancy was registered in 31.5% (161) of cases. Maternal age as a risk factor was recorded in 22.5% (115) of ANC cards: adolescent pregnancy ($\leq$16 years old) in 5.5% (28) and 17% (87) higher than 35 years old. Thirty percent (155) had a prior history of miscarriages, and 3.9% (20) had stillbirths, although no more details were specified in their ANC cards. A rate of 10% was found for grandmultiparous women.

**ANC: Screenings, maternal diagnosis, and treatments.** Table 1 describes the screenings registered in the ANC pregnancy cards for the study participants (n = 511) and according to the number of ANC contacts: inadequate (1–3), adequate (4–7) and complete (8 and above). Lack of economic support in 36% (184) and lack of reagents in 2% (10) were the reasons registered in the ANC pregnancy cards for screening absence. Pregnant women had no screening

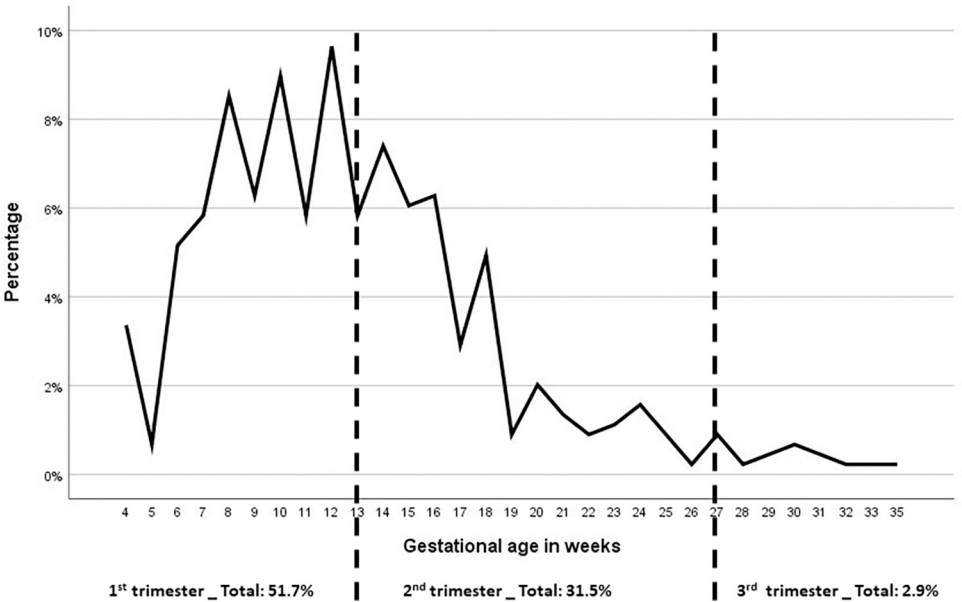

**Fig 1. Timing of first ANC contact according to gestational age in weeks and trimesters (data for 446 participants).** Distribution of gestational age in weeks and by trimester at the initiation of ANC contact, as follow: 51.7% (268/446) in the first trimester, 31.5% (163/446) during the second and 2.9% (15/446) in the third trimester. Data regarding the timing of the first ANC contact were missing from 65 (14.6%) ANC pregnancy cards.

done for HIV in 4.5%, syphilis in 8.8%, HBV 39.3%, malaria 25.8%, hemoglobin 24.5%, urine test 29.7%, blood glucose in 45.4%, stool exams 27.8% and 41.1% had no ultrasound.

Screening completion for blood group, HIV, malaria, urine, hemoglobin, and coproparasitological exam were found to have a statistically significant difference (p<0.001) for the complete ANC group when compared to the inadequate and adequate ANC groups as described in Table 1.

Free interventions, such as administration of one dose of sulfadoxine-pyrimethamine for intermittent preventive treatment for malaria during pregnancy (IPTp), were given to 47.4% (242) of women, 43% (220) received the two doses, and 9.6% (49) had none. Insecticide-treated nets were delivered to 88.5% (442) of the pregnant women. Regarding immunization, 94.5% (483) had received tetanus toxoid immunization as follows: 1.8% (9) had one dose, 30.5% (155) had two, 22.3% (114) had three, 17% (87) had four, and 23% (118) had the complete five doses. No dose was registered in 5.5% (28) of the ANC pregnancy cards.

The identified maternal ANC diagnosis and treatments prescribed are described in Table 2. Additional maternal problems identified during ANC follow-up were first-trimester bleeding in 0.2% (1), second trimester bleeding in 0.2% (1), and third trimester bleeding in 1.2% (6). All pregnant women with preeclampsia and bleeding were referred to the HAM maternity unit for further evaluation.

## Discussion

Neonatal mortality rate and morbidity due to adverse birth outcomes (ABOs) in Sao Tome & Principe remain high despite the recent significant reduction in maternal deaths [30–32]. Thus, we sought to analyze ANC through screenings performed and problems commonly detected among pregnant women in STP, since a key component to reduce neonatal morbidity and mortality is having a high-quality ANC [1–3].

**Table 1. Screening completion comparison according to the frequency of ANC contacts: Inadequate (1 to 3), adequate (4 to 7) and complete (≥8).**

| | Total n = 511 (%) | 1–3 ANC contacts n = 76 (14.9%) | 4 to 7 ANC contacts n = 237 (46.4%) | ≥8 ANC contacts n = 198 (38.7%) | p-value |
|---|---|---|---|---|---|
| **Blood group** | | | | | |
| yes | 333 (65.2) | 24 (31.6) | 152 (64.1) | 157 (79.3) | <0.001[#] |
| no | 178 (34.8) | 52 (68.4) | 85 (35.9) | 41 (20.7) | |
| **HIV** | | | | | |
| tested once | 213 (41.7) | 52 (68.4) | 104 (43.9) | 57 (28.8) | <0.001[#] |
| tested twice | 275 (53.8) | 20 (26.3) | 126 (53.2) | 129 (65.2) | |
| not tested | 23 (4.5) | 4 (5.3) | 7 (2.9) | 12 (6.0) | |
| **Syphilis** | | | | | |
| tested once | 460 (90.0) | 69 (90.8) | 212 (89.5) | 179 (90.4) | 0.717[*] |
| tested twice | 6 (1.2) | 0 | 2 (0.8) | 4 (2.0) | |
| not tested | 45 (8.8) | 7 (9.2) | 23 (9.7) | 15 (7.6) | |
| **Hepatitis B** | | | | | |
| tested | 310 (60.7) | 47 (61.8) | 143 (60.3) | 120 (60.6) | 0.870[*] |
| not tested | 201 (39.3) | 29 (38.2) | 94 (39.7) | 78 (39.4) | |
| **Malaria** | | | | | |
| tested once | 342 (66.9) | 24 (31.6) | 163 (68.8) | 155 (78.3) | <0.001[#] |
| tested twice | 37 (7.2) | 1 (1.3) | 14 (5.9) | 22 (11.1) | |
| not tested | 132 (25.8) | 51 (67.1) | 60 (25.3) | 21 (10.6) | |
| **Urine** | | | | | |
| tested once | 296 (58) | 27 (35.5) | 150 (63.3) | 119 (60.1) | <0.001[#] |
| tested twice | 63 (12.3) | 0 | 17 (7.2) | 46 (23.2) | |
| not tested | 152 (29.7) | 49 (64.5) | 70 (29.5) | 33 (16.7) | |
| **Coproparasitological exam** | | | | | |
| tested once | 329 (64.4) | 24 (31.6) | 164 (69.2) | 141 (71.2) | <0.001[#] |
| tested twice | 40 (7.8) | 0 | 8 (3.4) | 32 (16.2) | |
| not tested | 142 (27.8) | 52 (68.4) | 65 (27.4) | 25 (12.6) | |
| **Hemoglobin test** | | | | | |
| tested once | 296 (57.9) | 24 (31.6) | 158 (66.7) | 114 (57.6) | <0.001[#] |
| tested twice | 90 (17.6) | 2 (2.6) | 25 (10.5) | 63 (31.8) | |
| not tested | 125 (24.5) | 50 (65.8) | 54 (22.8) | 21 (10.6) | |
| **Ultrasound** | | | | | |
| 0 | 210 (41.1) | 60 (78.9) | 93 (39.2) | 57 (28.8) | <0.001[#] |
| 1 | 216 (42.3) | 13 (17.1) | 114 (48.2) | 89 (44.9) | |
| 2 | 72 (14.1) | 2 (2.6) | 23 (9.7) | 47 (23.7) | |
| ≥3 | 13 (2.5) | 1 (1.3) | 7 (2.9) | 5 (2.5) | |
| First trimester ultrasound | 161/216 (74.5) | 6/13 (46.1) | 66/114 (57.9) | 89/89 (100) | 0.005[#] |

[#]Pearson Chi-Square

[*]Fisher´s Exact Test

Antenatal care coverage in Sao Tome & Príncipe is a success compared to other SSA countries [8, 44]. Attendances are extremely high for a low-resource country. The minimum of eight ANC contacts, endorsed by the 2016 WHO´s guideline, was reached in almost forty percent. Moreover, 85.1% of the participants had four or more contacts when the rates reported in SSA for 4 visits were approximately 62% [44]. Regarding the timing of the first ANC booking, we also found a high percentage of pregnant women with early contact during the first

**Table 2. Total screenings performed, maternal ANC diagnosis and treatments prescribed for study participants (511).**

| ANC screenings | Total performed (at least once) (%) | Positive results/maternal ANC diagnosis (%) | Treatment administration (%) |
|---|---|---|---|
| HIV[†] | 488 (95.4) | 4 (0.8) | 4 (100) received antiretroviral |
| Syphilis[‡] | 466 (91.2) | 5 (1.0) | 3 (60) received ANC penicillin |
| Hepatitis B virus (HBV) surface antigen[†] | 310 (60.7) | 18 (5.8) | 6 (33.3) newborns received selective hepatitis B birth-dose vaccination (HepB-BD)[1] |
| Malaria[†] | 379 (74.2) | 3 (0.8) | 3 received artemisinin-based combination therapy |
| Blood group test[*] | 334 (65.3) | - | - |
| Rhesus factor for Rh negative[**] | 326 (63.8) | 19 (5.8) | No |
| Hemoglobin for anemia[***] | 386 (75.6) | 143 (37) | No |
| Blood glucose for hyperglicemia[****] | 279 (54.6) | 10 (3.6) | No |
| Urine for bacteriuria | 359 (70.2) | 155 (43.2) | 97 (62.5) received antibiotic: amoxicillin 92.7% (90/97) ampicillin 5.5% (5/97) cefotaxime 2% (2/97) |
| Coproparasitological exam for pathogenic intestinal parasite | 369 (72.2) | 220 (59.2) | 55 (25) |
| Vaginal smear for candida infection | 351 (68.7) | 104 (29.6) | 104 (100) treated with nitroimidazoles |
| Widal test for typhoid fever | 236 (46.2) | 35 (14.8) | 35 (100) were treated with amoxicillin |
| Sickling rapid laboratory test for sickle cell disease or trait | 284 (55.6) | 37 (13) | No |
| Blood pressure measurement[†] | 414 (81) | 37 (7.1) preeclampsia | 37 (100) |
| Obstetric ultrasound | 301 (58.9) | 15 twin pregnancies 2 major fetal malformations | - |

[†]free of costs screening tests

[1]While this study was being conducted, there was an expansion to universal newborn hepatitis B birth-dose vaccination (HepB-BD) without maternal screening. Before that, the country had a selective hepatitis B birth-dose vaccination (HepB-BD) strategy targeting infants born to mothers who tested positive for hepatitis B virus (HBV) surface antigen.

[*]17.4% (58) were group A, 26.3% (88) group B, 54.8% (183) group 0, and 1.5% (5) were group AB

[**]anti-D immunoglobulin is not available in STP

[***]Anemia was defined as a hemoglobin value less than 11 g/dL. ANC supplementation with iron/folate tablets was prescribed to 86.8% (444) of pregnant women, and no additional measure was described in the ANC pregnancy card for anemic mothers.

[****]Hyperglycemia was defined as glucose ≥105 mg/dL. No special measure or treatment was registered in these pregnant women ANC cards.

trimester. This finding of 51.7% is much higher than other studies in SSA, per example, Tanzania with 12.4%, Nigeria with 15.4%, Zambia with 17%, and Ethiopia with 27.5% [24, 45–47].

However, this is not equivalent to the provision of good quality ANC since most pregnant women in this study missed screenings, diagnosis, and treatments, which has an impact on neonatal health outcomes, highlighting an important ANC coverage-quality gap in Sao Tome & Principe [33]. For instance, anemia during pregnancy is a public health problem that leads to different life-threatening complications and poor pregnancy outcomes [13]. In this study, among those who were tested, 37% were anemic, similar to the 41.8% overall prevalence of maternal anemia in Africa [13]. Thus, anemia during pregnancy should be urgently addressed in STP to prevent related complications such as intrauterine growth retardation, prematurity, LBW, and fetal death [13].

Another maternal issue related to ABOs identified in this study is the high prevalence of asymptomatic urinary tract infections in pregnancy. From those who were able to perform a urine test, we identified a rate of 43.2% bacteriuria, which is more than double the range usually reported for pregnant women in SSA (2% to 16.43%) [14, 15, 47]. In addition to the high

prevalence of bacteriuria, we also identified that almost half of those in this situation go through all pregnancy without either receiving antibiotic treatment or repeating the urine test. Furthermore, and unlike culture (not available at STP), urinalysis fails to identify the etiologic agents and the antibiotic sensitivity pattern. Due to the above reasons, the real burden of bacteriuria in pregnant women and its associated neonatal complications (LBW and/or preterm birth) in STP are highly underestimated and need specific measures to address it.

Intestinal parasitic infections are known to be extremely frequent among children in STP [48], and in this study, we were able to identify that sixty percent of the pregnant women enrolled had at least one pathogenic intestinal parasite infection documented in their ANC pregnancy card. Anthelmintic treatment was administered in approximately one-quarter of pregnant women. Thus, considering this high prevalence as well as the burden of maternal anemia in the country, the implementation of preventive anthelminthic treatment for pregnant women after the first trimester as part of worm infection reduction programs should be discussed [6, 32].

Malaria was detected in three (0.8%) pregnant women, while prophylaxis with intermittent preventive treatment for malaria during pregnancy was provided to half of the women and a high proportion received an insecticide-treated net, similar to other nearby countries, such as Gabon [49].

HIV was detected in less than one percent of pregnant women, which is in accordance with published data (0.2%) for the country [49]. Syphilis was diagnosed in 1%, lower than the estimated pooled prevalence of syphilis of 2.87% for SSA [12]. This is a noteworthy point in the context of the triple elimination of HIV, syphilis and hepatitis B in Sao Tome & Principe [50–52].

Typhoid fever is endemic in STP, as it is common in SSA, which lacks access to clean water and adequate sanitation and found on 14.8% of pregnant women [40, 41]. However, the efficiency of the Widal test in diagnosing typhoid fever without other confirmatory tests is not of diagnostic value; thus, it should not be performed as a routine antenatal care practice in STP [53].

The low rate of gestational diabetes mellitus in this study can also be underestimating the reality, taking into consideration that half of the pregnant women only tested once for blood glucose. The number of preeclampsia cases detected was 7%, which is in accordance with the range of 3% to 10% published for SSA countries [54]. Another concern identified in this study is the risk for RhD alloimmunization, as the Rhesus factor was negative in 5.8% of pregnant women, similar to other nearby countries [16].

Sickle cell disease (SCD) or trait was suspected in 13% of pregnant women, highlighting a crucial need to understand the real burden of SCD in the country.

Regarding evidence-based screenings performed according to the number of ANC contacts, pregnant women with complete ANC had more screening completion in comparison to those with adequate and inadequate ANC, with a statistically significant difference (p<0.001). Nonetheless, having two hemoglobin and urine tests was only accomplished by thirty percent of those with complete ANC (≥8 contacts), illustrating that most pregnant women, even those with complete ANC, are not able to accomplish the minimum recommended screenings throughout the pregnancy.

It should be highlighted that all the above discussed antenatal diagnoses are underestimated since many pregnant women in this study did not test at all, and lack of economic support was the main reason documented. Pregnant women may not have the financial resources needed to pay for ANC screenings, especially if they cost 17 USD when living on a budget of 1.40 USD per day [32]. Missing out these evidence-based ANC screenings and interventions due to their high-costs represents losing the momentum for adequate antepartum intervention of maternal

conditions amenable to treatment, consequently missing key opportunities in the prevention of risk factors linked to perinatal mortality and ABOs [5, 6, 22]. Therefore, providing a health system in STP that broadens the benefits of free ANC screening can go a long way to improve long-term gains for maternal and newborn health [22, 24, 37, 47].

## Strengths and limitations

This study has the strength of being based on a large national sample. We included data from ANC pregnancy cards from 511 pregnant women, a similar sample used in the 2019 Multiple Indicator Cluster Survey (MICS) for Sao Tome & Principe from UNICEF [30]. Moreover, contrary to MICS, our study is not vulnerable to recall bias, as the mothers included had the ANC card with them and had just delivered the baby. Additionally, most of the studies in SSA only analyze ANC utilization in terms of the number and timing of attendances. Our study included routine screenings, maternal diagnosis, and treatments in addition to the number of contacts per pregnant woman. Moreover, this study was conducted on HAM, where approximately 82% of all births in the country take place; therefore, it also covered most rural areas.

Therefore, this study can be a model for other similar low-resource countries for detecting the main gaps in the country´s ANC service, enabling the design of target intervention and better allocation of resources to where they are needed.

However, there were some limitations that should be pointed out: proper records and quality of care given could not be assessed, and ANC practices may differ in terms of access to health care, health worker motivation and training, and availability of health service across the islands. The STP´s ANC pregnancy card was designed to make documentation complete and easier, but it was found that important information was not always adequately recorded by the nurses. Details of previous pregnancies, contraception, planned pregnancy, present medical problems and signaling of high-risk pregnancies were often missing. Height, weight, and body mass index are not registered during ANC visits in STP, as it is not mandatory according to the WHO 2016 recommendations, but this could be reviewed for this setting [6].

## Conclusion

Sao Tome & Principe have made remarkable progress toward ensuring that no pregnant woman is left without antenatal care. Nonetheless, most still miss evidence-based screenings, diagnosis, and treatment opportunities with an impact on neonatal health outcomes.

Strategies such as 1) screening all pregnant women for anemia and bacteriuria, 2) preventive administration of anthelmintic drugs after the first trimester, 3) implementation of universal access to anti-D immunoglobulin for Rh-negative pregnant women, 4) provision of novel inexpensive sickle cell disease point-of-care test kits in the country, and 5) implementation of a total free antenatal care health policy should be discussed due to the country profile and for enhancing neonatal health outcomes.

These measures would enhance the prompt detection of maternal problems amenable to ANC treatment, preventing ABOs and reducing neonatal mortality in STP.

## Supporting information

**S1 Text. Inclusivity in global research.**
(DOCX)

## Acknowledgments

A special remark for the late Professor João Luís Baptista PhD MD—AV research co-supervisor—a great man who was a thinker and a fighter for Africa's improvement of public health. We are indebted to all the women who participated in the study. The authors would like to thank Elizabeth Carvalho and the 1) medical team and nurses of Hospital Dr. Ayres de Menezes Maternity for their support, especially to the chief-nurse Paulina Oliveira, and 2) Ana Sequeira, Rita Coelho, Ana Margalha, Ana Castro, Alexandra Coelho and Inês Gomes for field support. We would like to acknowledge Instituto Camões, I.P. for the logistic support in STP.

## Author Contributions

**Conceptualization:** Alexandra Vasconcelos, Filomena Pereira, Maria Céu Machado.

**Data curation:** Alexandra Vasconcelos, Marta Alves.

**Formal analysis:** Alexandra Vasconcelos, Marta Alves, Ana Luísa Papoila.

**Funding acquisition:** Alexandra Vasconcelos.

**Investigation:** Alexandra Vasconcelos.

**Methodology:** Alexandra Vasconcelos, Marta Alves, Ana Luísa Papoila.

**Project administration:** Alexandra Vasconcelos.

**Resources:** Alexandra Vasconcelos, Swasilanne Sousa, Nelson Bandeira.

**Software:** Marta Alves, Ana Luísa Papoila.

**Supervision:** Filomena Pereira, Maria Céu Machado.

**Visualization:** Swasilanne Sousa, Nelson Bandeira.

**Writing – original draft:** Alexandra Vasconcelos.

**Writing – review & editing:** Alexandra Vasconcelos, Swasilanne Sousa, Nelson Bandeira, Marta Alves, Ana Luísa Papoila, Filomena Pereira, Maria Céu Machado.

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
