## [Decision Letter · Decision Letter 0]

25 Aug 2022

PGPH-D-22-00754

Antenatal screening tests and antenatal problems among pregnant women in Sao Tome and Principe: a hospital-based study

Dear Dr. Vasconcelos,

Thank you for submitting your manuscript to PLOS Global Public Health. After careful consideration, we feel that it has merit but does not fully meet PLOS Global Public Health’s publication criteria as it currently stands. Therefore, we invite you to submit a revised version of the manuscript that addresses the points raised during the review process.

Thanks for submitting your manuscript for consideration. A major revision is required prior to a final decision.

We look forward to receiving your revised manuscript.

Kind regards,

Charles Anawo Ameh, PhD

Academic Editor

Journal Requirements:

1. Please include a complete copy of PLOS’ questionnaire on inclusivity in global research in your revised manuscript. Our policy for research in this area aims to improve transparency in the reporting of research performed outside of researchers’ own country or community. The policy applies to researchers who have travelled to a different country to conduct research, research with Indigenous populations or their lands, and research on cultural artefacts. The questionnaire can also be requested at the journal’s discretion for any other submissions, even if these conditions are not met.  Please find more information on the policy and a link to download a blank copy of the questionnaire here: https://journals.plos.org/globalpublichealth/s/best-practices-in-research-reporting. Please upload a completed version of your questionnaire as Supporting Information when you resubmit your manuscript

2. In the online submission form, you indicated that "The datasets used and/or analyzed during the current study are available from the corresponding author on reasonable request.". All PLOS journals now require all data underlying the findings described in their manuscript to be freely available to other researchers, either 1. In a public repository, 2. Within the manuscript itself, or 3. Uploaded as supplementary information.

3. Please upload all main figures as separate Figure files in .tif or .eps format only and remove the embedded figures from the manuscript file. For more information about how to convert and format your figure files please see our guidelines: 

4. We do not publish any copyright or trademark symbols that usually accompany proprietary names, eg (R), (C), or TM  (e.g. next to drug or reagent names). Please remove all instances of trademark/copyright symbols throughout the text, including © on pages 12 and 15.

Additional Editor Comments (if provided):

Thanks for submitting your manuscript for consideration. A major revision is required prior to a final decision.

Reviewers' comments:

Reviewer's Responses to Questions

**Comments to the Author**

1. Does this manuscript meet PLOS Global Public Health’s publication criteria? Is the manuscript technically sound, and do the data support the conclusions? The manuscript must describe methodologically and ethically rigorous research with conclusions that are appropriately drawn based on the data presented.

Reviewer #1: Partly

Reviewer #2: Partly

2. Has the statistical analysis been performed appropriately and rigorously?

Reviewer #1: Yes

Reviewer #2: No

3. Have the authors made all data underlying the findings in their manuscript fully available (please refer to the Data Availability Statement at the start of the manuscript PDF file)?

Reviewer #1: Yes

Reviewer #2: Yes

4. Is the manuscript presented in an intelligible fashion and written in standard English?

Reviewer #1: Yes

Reviewer #2: Yes

5. Review Comments to the Author

Reviewer #1: The manuscript, entitled “Antenatal screening tests and antenatal problems among pregnant women in Sao Tome and Principe: a hospital-based study”, used a retrospective study design to review the ANC cards of 518 pregnant women. The purpose of the study was to identify screenings performed and antenatal problems with a focus on how to improve the ANC in STP to reach the newborn health post-2015 Sustainable Development Goals. The study found that the ANC screening service was less than optimal. Specifically, pregnant women had no screening done for HIV in 7.6%, syphilis in 10%, HBV 40.2%, malaria 27%, haemoglobin 37.5%, blood glucose 46.2%, urine test 31%, stool exams 29% and 42.1% had no ultrasound.

In my opinion, the study is worthwhile considering recent calls to focus on quality of care and not just coverage of essential indicators. The study shows the coverage-quality gap in STP indicators for maternal health services and confirms what has already been seen around the world about the coverage-quality gap

The manuscript is well written in standard English, and the methodology used in the study is sound, and the details are well elaborated. Nevertheless, there are some minor concerns that need to be addressed to make the manuscript scientifically suitable for publication in PLOS Global Public Health. The concerns are outlined below according to the sections:

General comments

Authors should ensure that the manuscript is proofread and issues of grammar and wrong spelling are rectified. In addition, there should be consistency in the English used, either American or British, but not both, which is obvious in the manuscript.

Introduction

Line 58-60. There is a problem with the sentence structure. The phrase “should be provided” has to be deleted for the sentence to flow well.

Line 68- 70. Correct the pronoun “it” in the sentence to reflect the use of the pleural form of “infections” [in pregnancy]. A similar issue appears in line 78-79 where “go” is used for the singular form of “sickle cell disease”

Line 89-91. I don’t think it is true that STP lacks skilled medical personnel and health infrastructure. They may be present but woefully inadequate. Indeed, lack and inadequate are not the same, and it is unlikely that it is the case of the forma in STP. Clearly, the current study was health facility based, and that alone is enough to disqualify the use of “lack”. Furthermore, a hospital is regarded as a health infrastructure, and in this manuscript, it is indicated that the majority of births take place in the only hospital in STP.

Line 100. There is an issue with grammar. “Studies” is pleural. Therefore, check the use of “analyses”. It is better to use most studies… analyze or analyse this service…, instead of analyses.

Line 103. The WHO document on positive pregnancy experience is referred to in the manuscript. Authors should, therefore, comply and use contacts and not visits as recommended. That should be corrected throughout the manuscript

Materials and methods

Line 114-115. Correct sentence structure.

Authors should also acknowledge that the study was purely descriptive in nature as part of the study design.

Line 171-172. Refer to my earlier comment on the use of ANC visits.

It is conceptually appropriate to place the data source section before the variable section.

Nowhere in the methods section has it been mentioned what was used to extract the information for the ANC record cards. Then a questionnaire is mentioned towards the end under data management. What was this questionnaire used for?

Line 217. What does it mean to put percentages in parenthesis after the mention of frequencies? The two are obviously not the same and do not provide the same information.

Authors indicated that anonymity and the safety of participants were ensured. How was safety ensured in the study?

Results

Line 239-241. The sentence is confusing? Is the mean age and standard deviation for the maternal age or the age of the pregnancy? The range shows a maximum of 43 years. I don’t think a pregnancy can last that long. The following information shows the information is about age, so I wonder why “current pregnancy” was mentioned in the beginning sentence.

The variable employment was described in the methods section as employed and not employed. In the results, there is another category—students. Does it mean that students are not employed? If so, why are they not added to the unemployed category or the employed if the respondent is a student and is involved in an income-generating activity?

Here again, visits have been used instead of contacts.

Figure 1 does not provide the information described in the narrative. I can see that the maximum value (percentage) from the figure is 10.0%. How did the authors come up with the 51.7% for the first trimester? What was the description used for the trimesters? It is only stated in the methods that information on the timing of the first contact was collected. The figure should either be compressed into the trimesters or the narrative should capture the weeks. Data labels could also enhance readability.

Tables 3, 4, and 5 are confusing. The titles mention “adequacy of use”. Is there any standards for the coverage? What are the criteria for measuring adequacy in this case? Appropriate titles should be given to disease tables to reflect the information they provide.

Also, are the data being compared in these tables? For example, are the authors comparing the coverage of HIV testing in the 1st test group with the 2nd test group? If comparisons are being made, which appears to be the case, the statistical test needs to be mentioned.

Discussion

Since 2016, the WHO has recommended at least 8 ANC contacts. In the second paragraph of the discussion, the authors said that the minimum number of contacts was four. Is it the case in STP that the new recommendations have not been adopted?

Line 336-337. I spot a grammar issue in the sentence that needs to be corrected

Line 339. What does reference no. 8 serve in this situation?

Line 344-345. In this context, authors cannot use both either and neither. Please revise the sentence. Either is used with “or” and neither with “nor”.

Line 410. Is “testes” the word?

The discussion is unnecessarily long. I recognize that many interventions were examined, and it is tempting to report and discuss all these interventions. However, the authors need to bring out the main message of the results and their implications for quality care.

Conclusion

Provide a conclusion that is succinct and supported by the current findings

Reviewer #2: The authors have done well to write the methods in detail but the study design and sampling method was not clearly explained. It would be good to give the total population size and proportion of female population to understand some of the statistics in a better way. Though it was a chart review study but the way the source documents were obtained is unclear. The tables need improvement. Statistical tests could also be done for group-wise comparison (e.g according to number of visits). The subject may be locally relevant.

6. PLOS authors have the option to publish the peer review history of their article (what does this mean?). If published, this will include your full peer review and any attached files.

**Do you want your identity to be public for this peer review?** For information about this choice, including consent withdrawal, please see our Privacy Policy.

Reviewer #1: **Yes: **Michael Boah (PhD)

Reviewer #2: No

---

## [Decision Letter · Decision Letter 1]

23 Nov 2022

PGPH-D-22-00754R1

Antenatal screenings and maternal diagnosis among pregnant women in Sao Tome & Principe - missed opportunities to improve neonatal health: a hospital-based study

Dear Dr. Vasconcelos,

Thank you for submitting your manuscript to PLOS Global Public Health. After careful consideration, we feel that it has merit but does not fully meet PLOS Global Public Health’s publication criteria as it currently stands. Therefore, we invite you to submit a revised version of the manuscript that addresses the points raised during the review process.

Both of the previous reviewers have assessed the revised manuscript, and are overall happy with the changes. Reviewer 2 has suggested limiting the discussion somewhat. Whilst there is no word limit for these sections at PLOS Global Public Health, we wanted to invite you to address these suggestions, should you wish to do so. Once you are happy with the manuscript, please resubmit your manuscript.

We look forward to receiving your revised manuscript.

Kind regards,

Hanna Landenmark

Staff Editor

Journal Requirements:

b. If any authors received a salary from any of your funders, please state which authors and which funders.

Additional Editor Comments (if provided):

Reviewers' comments:

Reviewer's Responses to Questions

**Comments to the Author**

1. If the authors have adequately addressed your comments raised in a previous round of review and you feel that this manuscript is now acceptable for publication, you may indicate that here to bypass the “Comments to the Author” section, enter your conflict of interest statement in the “Confidential to Editor” section, and submit your "Accept" recommendation.

Reviewer #1: All comments have been addressed

Reviewer #2: (No Response)

2. Does this manuscript meet PLOS Global Public Health’s publication criteria? Is the manuscript technically sound, and do the data support the conclusions? The manuscript must describe methodologically and ethically rigorous research with conclusions that are appropriately drawn based on the data presented.

Reviewer #1: Yes

Reviewer #2: Yes

3. Has the statistical analysis been performed appropriately and rigorously?

Reviewer #1: Yes

Reviewer #2: Yes

4. Have the authors made all data underlying the findings in their manuscript fully available (please refer to the Data Availability Statement at the start of the manuscript PDF file)?

Reviewer #1: Yes

Reviewer #2: Yes

5. Is the manuscript presented in an intelligible fashion and written in standard English?

Reviewer #1: Yes

Reviewer #2: Yes

6. Review Comments to the Author

Reviewer #1: Thank you for your time in addressing the comments raised. I deem the revised manuscript scientifically suitable for consideration for publication in PlOS Global Public Health

Reviewer #2: The authors have revised the manuscript to align with the comments from all reviewers. However, discussion is still quite lengthy. The authors are advised to limit discussing all the factors and retain only for the most important ones and keep one paragraph to discuss the rest. Ofcourse, the authors will have to be mindful of the local context but a revision to shorten the text will be better.

7. PLOS authors have the option to publish the peer review history of their article (what does this mean?). If published, this will include your full peer review and any attached files.

**Do you want your identity to be public for this peer review?** For information about this choice, including consent withdrawal, please see our Privacy Policy.

Reviewer #1: **Yes: **Dr Michael Boah

Reviewer #2: No

---

## [Editor Report · Decision Letter 2]

8 Dec 2022

Antenatal screenings and maternal diagnosis among pregnant women in Sao Tome & Principe - missed opportunities to improve neonatal health: a hospital-based study

PGPH-D-22-00754R2

Dear Dra. Vasconcelos,

We are pleased to inform you that your manuscript 'Antenatal screenings and maternal diagnosis among pregnant women in Sao Tome & Principe - missed opportunities to improve neonatal health: a hospital-based study' has been provisionally accepted for publication in PLOS Global Public Health.

Best regards,

Julia Robinson

Executive Editor